# Genome-Wide Association Analysis of Senescence-Related Traits in Maize

**DOI:** 10.3390/ijms232415897

**Published:** 2022-12-14

**Authors:** Venkata Rami Reddy Yannam, Marlon Caicedo, Rosa Ana Malvar, Bernardo Ordás

**Affiliations:** 1Mision Biológica de Galicia, Spanish National Research Council (CSIC), 36001 Pontevedra, Spain; 2Sustainable Field Crops Programme, IRTA (Institute for Food and Agricultural Research and Technology), 25198 Lleida, Spain; 3Estación Experimental Tropical Pichilingue, Programa de Maíz, Instituto Nacional de Investigaciones Agropecuarias (INIAP), Quito 170518, Ecuador

**Keywords:** maize, senescence, genome-wide association study (GWAS), QTL, candidate gene

## Abstract

Senescence is a programmed process that involves the destruction of the photosynthesis apparatus and the relocation of nutrients to the grain. Identifying senescence-associated genes is essential to adapting varieties for the duration of the cultivation cycle. A genome-wide association study (GWAS) was performed using 400 inbred maize lines with 156,164 SNPs to study the genetic architecture of senescence-related traits and their relationship with agronomic traits. We estimated the timing of senescence to be 45 days after anthesis in the whole plant and specifically in the husks. A list of genes identified in a previous RNAseq experiment as involved in senescence (core senescence genes) was used to propose candidate genes in the vicinity of the significant SNPs. Forty-six QTLs of moderate to high effect were found for senescence traits, including specific QTLs for husk senescence. The allele that delayed senescence primarily increased grain yield and moisture. Seven and one significant SNPs were found in the coding and promoter regions of eight core senescence genes, respectively. These genes could be potential candidates for generating a new variation by genome editing for functional analysis and breeding purposes, particularly Zm00001d014796, which could be responsible for a QTL of senescence found in multiple studies.

## 1. Introduction

The two challenges facing the world today are food and energy. In the present situation, the most substantial proportion of energy originates from non-renewable energy sources, which are not sustainable and are harmful to the environment because of the high CO_2_ outflows that cause the greenhouse effect or global warming [1]. Utilizing plant biomass for bioenergy is an economical way of decreasing CO_2_ emissions. Agricultural residues are an interesting and attractive bioenergy feedstock since they do not rival food, and their exploitation does not entail arable land loss. In the case of cereals, dual-purpose varieties can be exploited using the grain for feed or food, and the stalks and cobs for bioenergy. More experimental information about the genetic basis of traits related to the exploitation of residuals and the correlations between residual and grain characteristics is needed to design an efficient breeding strategy for developing dual-purpose varieties.

Maize (*Zea mays* L.) is an essential annual cereal crop for human food, animal feed, and industrial use. It has a C_4_ photosynthesis mechanism that is very efficient and generates a large amount of biomass. The life cycle of maize is classified into vegetative phases (sowing to flowering), grain filling, senescence, and death. Senescence is a complex activity that involves the degradation of the photosynthetic apparatus and many activities, such as the breakdown of macromolecules, accumulation, redistribution, and remobilization of nutrients, such as nitrogen [2]. There is a vertical variation profile of leaf senescence: the leaves situated in the middle, mainly surrounding the ears, remain green for a more extended period [3,4]. In many commercial maize hybrids, the husks turn dry before the remaining leaves, resulting in the appearance of all plant green, except the husk surrounding the ears. These varieties are named stay-green, in contrast to those with earlier senescence [5,6].

Genetic variations exist in the timing and rate of senescence [7,8]. Delays in senescence can increase carbon assimilation, resulting in higher biomass and grain yield. The duration of active photosynthesis in leaves is correlated with the production of plant residues in maize (leaves, stalks, and cobs) and, therefore, impacts the utilization of residuals as feedstock for bioenergy [9]. Dry matter accumulation increased up to 63% in the new hybrid era of maize released in the 20th century compared to earlier (1939–1999), partly due to the selection of late senescence (LS) hybrids [10]. However, the delay in senescence is also associated with increased stover moisture [1], which could be detrimental for transporting or storing agricultural residuals to produce bioenergy.

Knowledge of the genetic architecture behind the variation of leaf senescence facilitates the optimization of senescence characteristics for maize breeding, mainly grain and biomass production, moisture, and nitrogen and water use efficiency [1]. QTLs distributed along the genome have been generally found for senescence in maize through linkage and genome-wide association mapping [11,12,13]. In different experiments, senescence was measured by distinct methodologies, including visual estimation of green leaf area, measurement of chlorophyll content, or the maximum quantum efficiency of Photosystem II (PSII) (Fv/Fm), which could reflect different biological processes [14,15,16]. However, the decay of photosynthetic activity during senescence was not directly measured in previous QTL experiments.

The combination of GWAS and RNAseq provides a high resolution for disentangling the genetic basis of complex traits, and it is widely used as a standard approach [17,18,19]. RNAseq studies detect individual genes that are related to the trait. Still, GWAS detects genomic regions, not individual genes, which are related to the trait’s variation—that is, those important in artificial selection. In addition, GWAS allows quantification of the association between genomic regions and traits.

The primary general aim of this experiment was to study the genetic architecture of the duration of photosynthetic activity in maize leaves using GWAS. Within this wider purpose, a specific objective was to identify candidate genes for senescence traits by combining GWAS and RNAseq. A second specific objective was to evaluate the relationship between senescence and agronomic traits at the phenotypic and molecular levels. Finally, a third specific objective was to study the genetics of husk senescence independently of the remaining leaves.

## 2. Results

The Evanno test resulted in six subpopulations (k = 6) as the most significant cluster of subpopulations (ΔK) (Figure 1a). The proportion of the estimated membership of the sub-population is shown in Figure 1b. More than 40% of inbred lines (176 genotypes) were grouped in sub-population 3, considered a major group, and 77, 56, and 46 inbred lines were grouped in sub-populations 1, 4, and 5, respectively.

A summary of mean phenotypic traits and individual year data is shown in Table 1, and Appendix A, respectively, and normal distribution plots are presented in Appendix A. The sizable genotypic variation increases the possibility of detecting an association between markers and senescence traits. Heritability was moderate or high (0.6–0.9) for all traits except RM (0.14) (Table 1). The heritability of VSP was higher than the heritability of PA (0.85 vs. 0.65). Interestingly, the variance component due to the genotypic × year interaction was much lower than the genetic variance component in VSP but not in PA (Table 1).

Most of the lines with a VSP equal to 1 had low grain moisture and low yield (Figure 2). In contrast, most of the lines with a VSP of 4 were at the top of the graphic, with high values of grain moisture, and many of them with high grain yield. The lines with VSP 2 and 3 had intermediate grain yield and moisture values. Only four lines (NC288, AusTRCF305801, AusTRCF305802, T264) had an average VSP equal to 5, which had high moisture. There were several exceptions to the relationship between plant visual scale and grain moisture and yield. For example, one inbred line with a VSP equal to 1 had a relatively high yield, and another with a VSP equal to 3 had low moisture.

Almost all genotypic and phenotypic correlations among traits were significant (*p* > 0.05) (Figure 3). Regarding the genetic correlations between senescence traits, the significant genetic correlation between VSP and PA stood out. The genetic correlation between VSP and VSH was 0.78, which implies that about 40% of the genetic variation in VSH is not explained by its relationship with VSP. The genetic correlations between senescence and agronomic traits were moderate to high and positive. The correlation was higher between senescence and residual traits than between senescence and grain traits. The senescence trait with higher correlations with agronomic traits was VSP (RDW-0.82, RM-1.00, GDW-0.79, GM-0.74). In particular, there was a very high genetic correlation between VSP and RM.

### 2.1. Association between SNPs and Traits

We identified 46 markers significantly associated with senescence-related traits (VSP, VSH, PA, SC, PHVDIF) distributed in all chromosomes (Table 2 and Figure 4). The allelic effects ranged from 0.5 to 1.2 for VSP and VSH, while PHVDIF ranged from 0.4 to 0.7. For PA, the allelic effects varied from 3 to 6 mol CO_2_ m^−2^ s^−1^, and for SC, they varied from 0.01 to 0.04 mol H_2_O m^−2^ s^−1^. The proportion of phenotypic variance (R^2^) varied between 4 and 8% across all the traits. The number of marker trait associations (MTAs) was ten or close to ten for all senescence traits (VSP-8 MTAs, PA-8 MTAs, SC-9 MTAs, PHVDIF-8 MTAs), except for VSH, which doubled that number (22 MTAs). Eight markers were significant for more than one variable. Marker S6_163349566 on chromosome six was significant for visual and quantitative variables (VSP, VSH, PA). The sign of the allele effect was the same for the three traits; that is, the allele that increased VSP also increased VSH and PA. S1_209120152 was also associated with a visual scale of the husk (VSH) and a quantitative variable (PA). Again, the sign of the allele effect was coincident. S7_139201593 (C/T), S5_214537055 (A/G), S5_167009258 (C/T), and S3_5884845 (C/T) were significant for PA and SC. S5_147166110 (C/T) and S5_61802239 (C/T) were significant for the two visual traits, VSP and VSH. Although S6_162646831 is the only SNP significantly associated with senescence (VSP) *p* = 3.76 × 10^−5^, allelic effect = 0.67, and an agronomic trait (RDW) *p* = 3.40 × 10^−5^, allelic effect = 13.83), among all forty-six combinations of the senescence traits and the agronomic traits, the sign of the allelic effect was the same, except in three cases (S1_6423401, S2_184012260, S3_217840430) (Table 2). The alleles that delayed senescence mostly increased the weight and moisture of the grain and residuals.

### 2.2. Candidate Genes

We found sixty-one genes (two genes are repeated) in the window of ±200 kb of the significant SNPs (Appendix A). Among these, eight were included in the core senescence genes in the analysis of [13,20] and are the genes we propose as candidates (Table 3). Significant SNPs were located in the coding region in all candidate genes except S6_162646831, located in the promoter region, 119 bp far from the starting position. Zm00001d014796, Zm00001d017204, and Zm00001d014642 were upregulated, and Zm00001d005814, Zm00001d016802, Zm00001d038911, Zm00001d039155, and Zm00001d026501 were downregulated during senescence. The change in expression of all candidate genes, except Zm00001d005814, was consistent with the senescence of the lines in the study of [13,20] (Figure 5); that is, the up- or downregulation of those genes at senescence tended to occur earlier in the early senescence lines than in the late senescence lines. The expression of the gene Zm0000d014796 did not upregulate after flowering in the earliest senescence line NC292, at a difference from the other lines. However, the NC292 line already had a high expression level of this gene at flowering. In this line, the up- or downregulation of some senescence genes started much earlier than in the other lines, even at flowering.

### 2.3. Genomic Prediction Accuracies

The mean of 500 iterations of the genomic selection accuracies for the traits based on visual scales, VSP and VSH, and PHVDIF, were 0.24, 0.20, and 0.15, respectively, while the accuracies for the traits measured with the IRGA, PA, and SC were 0.14 and 0.08. The accuracies of the iterations followed a normal distribution (boxplots are shown in Appendix A).

## 3. Discussion

Senescence in annual crops, such as maize, involves changes in the expression of thousands of genes and the activation and inactivation of multiple metabolic pathways [9]. At the level of leaves, senescence involves the inactivation of photosynthesis, the activation of catabolic processes, and the transport of nutrients to developing kernels. One of the most apparent effects of foliar senescence is the breakdown of chlorophyll accompanied by a green color loss. Genotypes that remain green for a longer duration are named stay-green. If the photosynthesis activity is also functional for a more extended period, it is concluded that the stay-green is functional; if not, it is cosmetic [21]. We measured the progress of senescence 45 days after flowering using a visual scale (VSP trait) and found genetic variation for this trait. However, the VSP only measures the visual rating of leaf color, not photosynthesis activity. We also measured the rate of photosynthesis activity through the CO_2_ interchange per leaf area (PA trait) [22], and again, we found genetic variation.

A previous study observed that the visual color rating was correlated with chlorophyll content per unit area and that chlorophyll content was nearly correlated with photosynthesis activity [23]. Similarly, we found a very high genetic correlation between VSP and PA, which indicates the presence of photosynthetic activity when the genotype is green. Thus, we conclude that stay-green is mainly functional, not cosmetic, in maize, in agreement with [20]. The heritability for PA and VSP was moderate and high, respectively [24], indicating that these traits are easily changed by selection [25]. The differences in heritabilities among the two traits are due to a higher VG and, especially, a lower VGE detected in VSP compared to PA. If VG and VGE are relativized to VE, that is, divided by VE, the VG of VSP is twice the value of PA, and the VGE of VSP is five times lower than PA. With a visual scale, a general assessment of all plants, including all leaves, of a plot is carried out swiftly. In contrast, only a small area of one leaf from two plants per plot is analyzed with the infrared gas analyzer machine due to the limitation imposed by the relatively long time required for each measurement. The difference in the number of leaves and plants evaluated per plot could explain the differences in variance components between the two traits. Therefore, to alter the duration of the photosynthesis activity in maize, it is suggested to use a visual scale that has higher heritability and is economical and quick. Moreover, VSP had the highest genetic correlations with agronomic traits, and we expect an effective indirect response to agronomic traits when selected by VSP. Another possibility is that the infrared machine, which has more precision in individual measurements, is more capable of detecting the genotype × year interaction. This possibility deserves to be tested in a further experiment with fewer genotypes. If this is confirmed, infrared phenotyping can be used to integrate the genotype × year effect in the selection decision.

Although agronomical practices, such as row direction and plant spacing, have been studied for husk senescence [26], there is scarce literature on the genetic effects on husk senescence. We found a relatively high heritability, which indicates a substantial genetic component. The genetic correlation between VSH and VSP was moderate to high, but there was still a substantial part of the variation in VSH that VSP does not explain. This suggests that there are genetic mechanisms specific to the husks. We also studied the genetic basis of the difference between the senescence of the whole plant, mainly standard leaves, and husks, because there is some preference for hybrids with green leaves and dried husks at maturity. In most genotypes, the husks had earlier senescence than standard leaves, although they developed later, while in a few genotypes, senescence was simultaneous in standard leaves and husks. The differential senescence in standard leaves and husks requires coordination, which probably involves transcription factors whose activation varies across genotypes [27].

The expected effect of extending photosynthesis activity is grain yield increment, and we congruently found a high positive correlation between senescence traits and grain weight. However, we also identified a high genetic correlation between senescence and grain moisture; the delaying of senescence has a detrimental effect on grain moisture, which is prejudicial for grain storage [28]. The optimum duration of photosynthesis activity or the initiation of senescence, or the rate of senescence, depends on the environment. Each environment has an optimum duration of photosynthesis, and precise knowledge of this will better adapt the varieties to the environment. This knowledge would complement other phenology traits as a flowering time to adapt the cycle of the varieties to the environment [29]. For instance, in an environment with a short life cycle, farmers may prefer genotypes showing early senescence and vice versa in a long life cycle. The general pattern of higher moisture and yield with late senescence was evident. Still, we also identified lines with higher or lower yields or moisture than expected by the timing or rate of senescence. More research is needed to understand the relationship between senescence, agronomic traits, and environments.

We expected, a priori, a substantial effect of husk senescence on agronomic traits, particularly grain moisture, because the husks embrace the ears [6]. However, our data do not corroborate this hypothesis, as we found that the genetic correlation of grain moisture was even slightly higher with the senescence of the whole plant than with the senescence of the husks. Therefore, to change grain moisture or yield to adapt the crop cycle to the duration of the cultivation season, we recommend using a visual scale that considers the senescence of the whole plant. Modification of the senescence of the husk could still be important in nitrogen translocation during grain filling, as husk leaves play a crucial part in nitrogen translocation during grain filling [5,6].

Regarding cultivating varieties for dual purposes (grain for feed and residuals for bioenergy), we found that senescence traits are highly correlated to both the yield and the moisture of the residuals. The senescence affects the yield of grains and residuals in the same direction; therefore, there are no complications in improving both traits simultaneously. However, the adequate moisture of residuals depends on bioenergetic use [30,31]. For instance, in biogas production, the moisture content in residuals should be high, whereas in the case of combustion products, a low moisture content is preferable. The use of residuals for biogas is common in Atlantic Europe; the moisture of the grain should remain low, but the moisture of the residuals should be high [32]. These two opposite traits could be challenging to improve because senescence modifies both traits in the same direction as the high positive correlations shown in our experiment. Some specific lines that we identify could be an exception to this pattern and are valuable for developing varieties for dual purposes.

We did not find a large number of QTLs for plant senescence measured with either the CO_2_ interchange or the visual scale. A high power to detect QTLs for senescence is expected due to its high heritability [25]. Moreover, we used a liberal threshold that is expected to decrease Type II errors and reduce the number of false negatives. The number of QTLs we detected for senescence was similar to the number found in other senescence experiments [33,34,35] and lower than the number found in other traits of high heritability, such as flowering time [33,34,35]. The sizes of the allelic effects were quite large for the senescence traits; for example, for VSP, the effects ranged between 0.6 and 1.2, while the trait means and range were 2.5 and 4.0, respectively. Thus, a substantial change in the trait is expected by substituting one of the VSP alleles with the alternative allele. The relatively high allelic effects are consistent with a moderate to high percentage of variance explained by individual QTLs. The variance explained by individual QTLs related to senescence in this study fell inside the wide range of values (1–10%) found in other QTL stay-green experiments with temperate maize [33,34,35]. Thus, our and previous data are consistent with a polygenic inheritance for senescence, but the trait is probably not as highly quantitative as, for example, flowering time. The senescence’s genetic architecture also differs from the standard quantitative model, in which QTLs are not evenly distributed along the genome. Thus, most QTL experiments, including ours, consistently found more QTLs for senescence in chromosomes 1 and/or 5 [11,33,34,35,36], despite the different genetic constitution and type of the mapping populations, the experimental designs, or even the ways of measuring senescence.

We detected QTLs related to husk senescence for the first time, and some of them did not colocalize with QTLs for whole plant senescence, suggesting that they can be specific to husk senescence. This is consistent with a substantial part of the genetic variation of VSH not explained by its relation to VSP. For some of the significant SNPs, the allelic effects were high, for example, S1_6315814 (allelic effect = 1.16) or S3_217840430 (allelic effect = 1.04). Although VSH had lower heritability than VSP, the number of QTLs detected for VSH doubled the number detected for VSP, suggesting a more complex genetic architecture. We also detected QTLs for the difference between the senescence of the whole plant and husks (PHVDIF), some of them with a relatively high allelic effect, for example, S3_148031626 (allelic effect = 0.68). The QTLs for VSH or PHVDIF with a relatively significant effect could be useful in breeding for modifying the traits or investigating the genetic basis of senescence in husks.

We expect that several QTLs for senescence traits colocalize with QTLs for agronomic traits because the genetic correlation between the two types of traits is high. This was not the case, and only in one region were QTLs for the two types of traits localized together. However, out of 46 senescence QTLs, 43 had alleles with signs consistent with agronomic traits. The allele that had later senescence also had a higher yield and moisture, although the difference among the alleles for agronomic traits was not significant. The high consistency of the signs suggests a true association among several QTLs for both kinds of traits. However, we cannot have high confidence in any specific association, except one, because of the low probability associated with agronomic QTLs. The relatively large size of our mapping population has two opposite effects on the estimation of the QTL effects and years: an increment in precision due to the higher number of recombinants but a decrement in precision due to a higher experimental error. The experimental error could also explain the lack of strong significance of the senescence QTLs, despite the relatively large effects. The larger sizes of the field experiments limit the rise in precision achieved by increasing the number of genotypes in the association panels. Nowadays, when the limitation of insufficient genome coverage with molecular markers seems to be overcome, increasing the precision of phenotyping in field experiments is an active area of research [37,38].

The accuracy of genomic selection was higher for VSP than for PA, which was expected given its high heritability. The magnitude of the accuracy of genomic selection for VSP was in the range of traits, such as grain yield in maize [25]. Therefore, selection based only on molecular markers could be effective for senescence-related traits. The efficiency of phenotypic selection relative to genomic selection is the heritability divided by the prediction accuracy [25]. For VSP, phenotypic selection would be more than 3 times more effective than genomic selection. The low efficiency of the genomic selection in spite of the high heritability could be due to the great variability of the lines of the panel, which probably reduces the relatedness of the genotypes of the training and testing populations, and this decreases the prediction ability [39].

Eight significant SNPs were located in the coding or promoter regions of core senescence genes identified in a previous RNAseq experiment [13,20]. We propose these genes as candidates potentially involved in senescence and with polymorphisms that generate variability for senescence that could be useful in maize breeding. The functions of the eight genes are diverse. One of the candidate genes, Zm00001d026501, codifies the plastidic Glutamine synthetase2 enzyme (GS2), which is involved in the assimilation of photorespiratory ammonium [40]. An increase in nitrogen assimilation occurs when it is overexpressed, resulting in a high rate of photosynthesis [41].

On the other hand, a knockout mutation of GS2 in *Arabidopsis thaliana* shows a high ammonium content with a chlorotic phenotype [42]. In *Nicotiana tabacum* during senescence, GS2 is a pivotal enzyme that relocates the assimilation of ammonia from the chloroplast to the cytosol of mesophyll cells (MC) [43]. An experiment conducted with late senescence and an early senescence hybrid of maize found that GS2 was downregulated during senescence. The gene expression was retained longer in the late senescence hybrid than in the early senescing hybrid [44]. We also found that GS2 is downregulated at senescence, and that in late senescence lines, the downregulation of this gene tends to be later than in the early senescence lines (Figure 5a) [13,20].

Zm00001d014642 is orthologous to AT4G31540 of *A. thaliana* and is described as an EXCYST subunit or EXO70G1. It belongs to a gene family involved in diverse land plant functions: protein recycling, cytokinesis, biotic stress interactions, defense, secondary cell wall pathways, and many more [45]. The EXO70 subgroup participates in the regulation of leaf senescence by increasing its expression (Figure 5c), and it is also involved in other processes, such as nitrogen mobilization and the regulation of leaf senescence in wheat and soybean [46,47,48]. Additionally, the suppression or silencing of this gene leads to accelerated senescence in soybean [47]. Mutations in *A. thaliana* have fewer anthocyanins, which are essential components that safeguard the photosynthetic apparatus during senescence [49].

Of the eight significant SNPs located in the core senescence genes, only S6_162646831 was in the promoter region of Zm00001d038911. S6_162646831 was the only SNP significant for senescence and agronomic traits (VSP and RDW). This gene codifies a non-specific lipid transfer protein (nsLTP). The gene has an 85% similarity with its orthologue in sorghum, indicating a high level of conservation among both species. LTPs are small, essential proteins available in high amounts in higher plants and are responsible for the inter-transportation of phospholipids, glycolipids, fatty acids, and many other amphiphilic compounds between membranes in in vitro conditions [50,51,52]. Additionally, nsLTPs participate in signaling in plant defense against environmental stress, such as salt and cold stress [53,54,55]. However, the role of nsLTP in senescence is not known precisely.

The candidate gene Zm00001d017204 codifies an adenine phosphoribosyltransferase (APRT) enzyme that removes adenine and adenosine, which are considered toxic within chloroplasts [56]. In *A. thaliana*, APRT is a crucial metabolic enzyme in the inactivation of cytokinins, essential hormones that regulate the senescence process [57]. Zm00001d014796 is a nuclear export protein (NEP) interacting protein with a RING finger domain with a high degree of coincidence (85% identity) with similar genes in wheat and rice. In wheat, several genes that belong to this family change their expression during senescence [25]. This gene is located in one of the hotspots for QTL senescence at chromosome 5 [33,34,35,36] and could be one of the genes behind the QTLs consistently detected in the region, for example, the QTL detected by [33], which is 2 Mb apart from the candidate gene. Moreover, the gene regulation was consistent with the senescence of the lines in the RNAseq experiment. The earliest senescence line started very soon after its upregulation, which is indicative of the relevance of the gene, as it is probably involved in the initiation of senescence. The functions of the remaining three candidate genes have not yet been fully described. Zm00001d039155 is not described in maize, although it is orthologous in *A. thaliana* and is involved in histone deacetylation and condensation of heterochromatin. Zm00001d016802, which encodes ascorbate peroxidase (APX). APX is an enzymatic antioxidant that prevents the uncontrolled oxidation of cellular compounds by scavenging H_2_O_2_, one of the ROS in plants produced during photosynthesis and photorespiration [58,59]. However, Zm00001d016802 was downregulated in the previous RNAseq experiment (Figure 5e) [13,20], but we would expect that genes involved in antioxidant synthesis increase its expression at senescence, not the opposite. Finally, we found a significant SNP in Zm00001d005814, described as “Photosystem I chlorophyll a/b-binding protein 6, chloroplastic,” and it is one of the genes involved in light-harvest complex I (LHCI). This gene was downregulated during senescence, as expected from a gene involved in photosynthesis.

## 4. Material and Methods

### 4.1. Plant Material and Experimental Design

We evaluated 400 inbred lines of temperate, tropical/subtropical, and mixed origin, including representatives of the important groups used in breeding within temperate germplasm, such as Stiff Stalk, Lancaster, Iodent, and European Flint (Appendix A). The germplasm pedigree relationship and genetic diversity have been described in [60]. The inbred lines were provided by the North Central Regional Plant Introduction Station of the USA (NCRPIS). The inbred lines were evaluated at Misión Biológica de Galicia research station in Pontevedra, Spain, for senescence traits over three years (2017, 2018, and 2019) and biomass traits over two years (2018 and 2019). The soil was a sandy loam, and agronomic practices were the standard in the region. The inbred panel was evaluated using an augmented design with 17 blocks and six checks (A619, A632, A662, A665, PH207, and EP42) adapted to the experimental field conditions. Each experimental plot area was 1.9 m^2^, with a density of approximately 7 plants/m^2^. Each experimental plot consisted of 14 plants.

The genotypic data of the inbred lines used for association analysis were provided by Cornell University and consisted of 1,000,000 SNPs and are available on a public database, Panzea (http://www.panzea.org (accessed on 15 November 2018)) [60,61]. Heterozygous genotypes and insertion/deletion polymorphisms (INDELs) in this dataset were regarded as missing data. In total, 156,164 SNPs distributed over the entire maize genome were kept after excluding SNPs with more than 20% missing genotypic data and minor allele frequency (MAF) less than 0.05 in TASSEL 5.2.54 [62]. The physical position of the makers was provided according to version 4 of the Maize B73 RefGen_v4 (www.maizegdb.org (accessed on 20 of December 2018)) [63,64].

### 4.2. Population Structure Analysis

The genetic structure of the population was estimated using the Bayesian clustering algorithm implemented in the STRUCTURE software v2.3.4 [65] using an admixture model with burn-in and Monte Carlo Markov chain for 10,000 and 100,000 cycles, respectively. An uninterrupted series of K was tested from 1 to 10 in seven independent runs. The Evanno method [66] was used to calculate the most likely number of subpopulations with the STRUCTURE HARVESTER software [67].

### 4.3. Phenotypic Data

The net CO_2_ assimilation rate (mol CO_2_ m^−2^ s^−1^), which indicates photosynthesis activity (PA), was measured with an infrared gas analyzer (IRGA) portable photosynthesis system (LI-COR 6400XT, Lincoln, NE, USA) [68]. Measurements were taken in the middle part of the ear leaf of two plants randomly selected within each plot at 45 days after anthesis (DAA) [20]. We chose this specific moment based on previous experiments in which we measured senescence at regular intervals from anthesis to 90 DAA [20]. In those experiments, we found that the values of PA at flowering were similar among the genotypes and remained so until 45–60 DAA, when there were large differences in PA among genotypes due to differences in the timing or rate of senescence. Later, PA was negligible in most genotypes because senescence was completed in all of them. We concluded that 45 DAA is the optimum moment for detecting differences among genotypes in the timing and/or rate of senescence. Simultaneously with PA, stomatal conductance (SC) (mol H_2_O m^−2^s^−1^) was recorded with the portable photosynthesis system. After 45 DAA, the visual senescence of the plant (VSP) and the visual senescence of husks (VSH) were estimated on a visual scale from 1 to 5, where one corresponds to dried leaves and five to whole green leaves or husks. The visual scales were evaluated based on all the plants in the plots. We calculated the difference between VSP and VSH (PHVDIF) to assess the degree of association between the senescence of husks and the remaining parts of the plant.

A digital moisture meter calculated the grain moisture (GM) (g/Kg). The ears were removed from the vegetative fraction, and the grain dry weight (GDW) (g/plant) was estimated by fresh weight after adjustment by GM. Residual dry weight (RDW) (g/plant), including leaves and stalks, was measured after placing the fresh samples in a stove for five days at 60 °C. The residual moisture (RM) (g/Kg) was calculated by subtracting the proportion of dry weight from the freshly harvested plant weight.

### 4.4. Statistical Analysis

Analyses of variance and box and normal distribution plots were calculated for each individual year and for the combination of the three years using PROC MIXED and PROC UNIVARIATE procedures of SAS [69,70]. The phenotypic and genotypic correlation analysis was performed with three-year phenotypic values using Meta-R software developed by CIMMYT [71,72]. Heritability was estimated from variance components as:$$h^{2}=\frac{\sigma_{g}^{2}}{\sigma_{g}^{2}+\frac{\sigma_{gy}^{2}}{y}+\frac{\sigma_{g}^{2}}{ry}}$$
where $\sigma_{g}^{2}$ is the variance of genotypes, $\sigma_{gy}^{2}$ is the interaction of genotype and year variance, y is the number of years, $\sigma_{y}$ is the residual variance, and r is the number of replicates per year (*r* = 1).

### 4.5. GWAS Analysis

BLUEs (best linear unbiased estimates) of genotypes were calculated using PROC MIXED. Years, the interaction of the years with genotypes, and blocks within years were considered random effects. We performed the GWAS in TASSEL V5.2.25 [62], following a mixed linear model [73]: y=Xβ+Zu+e where *y* is the vector of BLUEs, *β* is a vector of SNP marker fixed-effects parameters, *u* is a vector of random additive effects of inbred lines, *X* and *Z* represent matrix, and *e* is a vector of random residuals. The random inbred line effect was estimated as Var(u) = K $\sigma_{a}^{2}$, where K is the *n* × *n* matrix of the pairwise kinship coefficient and $\sigma_{a}^{2}$ is the estimated additive genetic variance [73]. Restricted maximum likelihood estimates of variance components were obtained using the optimum compression level (compressed mixed linear model) and population parameters previously determined (P3D) in TASSEL. We used 10^−5^ as the threshold because the point where the observed and expected F-test statistics deviated in the Q-Q plot was equal to or lower than this value for all traits [20,74,75]. The significant associations were used to select candidate genes validated by an independent RNAseq experiment, as explained in the next section. Combining GWAS with RNA-seq decreases the false-positive rate and improves the accuracy of gene selection [76]. Circular Manhattan plots were generated by the CMplot package in R Software (https://github.com/YinLiLin/CMplot (accessed on 6 December 2021)). A QQ-plot was generated in TASSEL V5.2.25 [62].

### 4.6. Identification of Candidate Genes

We chose the genes within a window of ±200 kb of SNPs significantly associated with a senescence trait. Within those, we exclusively [20] considered as candidates the genes belonging to a list of core senescence genes identified in a previous study of RNAseq [13,20]. In the RNAseq experiment, the seven inbred lines were selected with identical flowering times but different durations of the period from flowering to complete senescence. NC292 and PHBB3 had early senescence, PHW79 and PHHB9 had late senescence, and PHT10, PHW52, and PA8627 had intermediate senescence. The leaf samples from each genotype were collected every 15 days from flowering to senescence to analyze RNAseq expression levels. Differential expression analysis was performed to compare the expression at flowering with the expression after flowering. In total, 1671 genes were considered core senescence genes because their change in expression at senescence, either up (1083 genes) or down (588 genes), was consistent across moments and lines. This list was used in the present work to select candidate genes, as previously noted. The RNAseq dataset is available publicly in the National Center for Biotechnology Information (NCBI) BioProject, https://www.ncbi.nlm.nih.gov/bioproject/ (accessed on 16 May 2022), PRJNA746402 [20]. The annotation and function of the candidate genes were done using the maize genome database (GDB) (https://www.maizegdb.org (accessed on 20 December 2018)) [63,64], uniport (https://www.uniprot.org (accessed on 2 September 2019)) annotation hub database [77], and a literature search of the gene and gene family.

### 4.7. Genomic Selection

Ridge regression best linear unbiased prediction (RR-BLUP) [78,79] was used to estimate marker effects and to develop prediction equations for genomic selection. We performed a cross-validation method to calculate the accuracies of the model using 80% of the population as training and 20% of the population as testing, with 500 iterations. The analyses were done in R software using the rrBLUP Package [80].

## 5. Conclusions

In this study, we presented the results of the evaluation of the largest association panel examined thus far in order to study the genetics of senescence, which was measured directly as photosynthesis decay for the first time in a mapping experiment in maize. The high genetic correlations (equal or higher than 0.8) and the high coincidence of the allelic effects (in 93% of the significant SNPs) among senescence and agronomic traits clearly showed the importance of senescence in adapting the crop cycle to the duration of the cultivation season. Eight of the significant SNPs detected in the association analysis were found in the coding or promoter regions of genes significantly associated with senescence in an independent RNAseq experiment. These genes are strong candidates for subsequent functional analyses to widen our knowledge of the genetics of plant senescence and for use in breeding to optimize the timing of senescence in different environments.

## Figures and Tables

**Figure 1 ijms-23-15897-f001:**
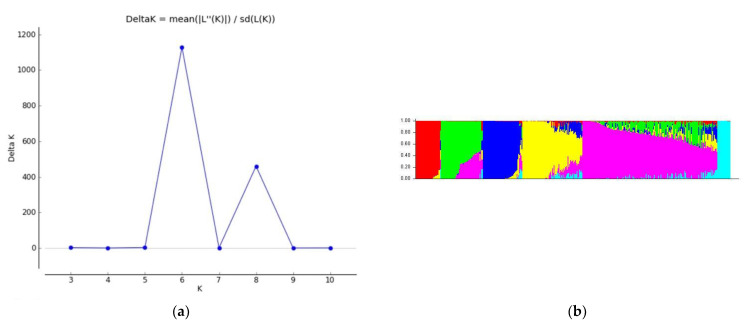
(**a**) Evaluation of STRUCTURE outputs; (**b**) estimated population structure. (**a**) The change in log probability of ΔK value (K); (**b**) the estimations of population structure at K = 6 for 400 inbred lines. Each inbred line is represented as a vertical segment, which is partitioned into 6 colored segments to denote estimated membership to the K cluster.

**Figure 2 ijms-23-15897-f002:**
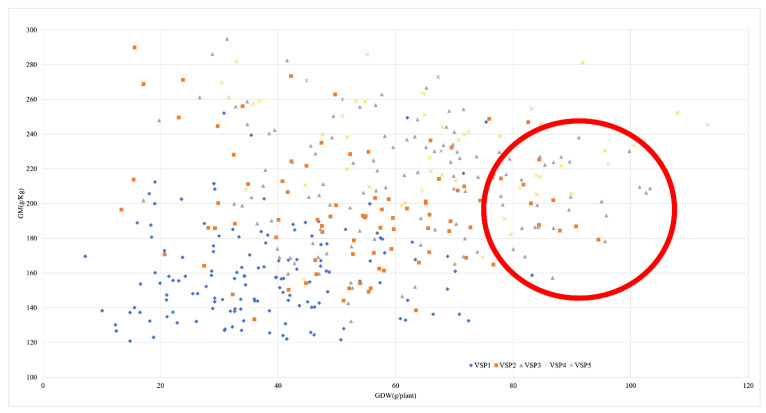
Comparison of early senescence and late senescence genotypes with agronomical traits. GM—grain moisture in grams/plant; GDW—grain dry weight in grams/plant; VSP1—genotypes correspond to a visual scale of plant value 1 (completely dry) at 45 DAA; VSP2—genotypes correspond to a visual scale of plant value 2 (partial green) at 45 DAA; VSP3—genotypes correspond to a visual scale of plant value 3 (intermediate green, intermediate dry) at 45 DAA; VSP4—genotypes corresponds to a visual scale of plant value 4 (partial dry) at 45 DAA; VSP5—genotypes corresponds to a visual scale of plant value 5 (complete green) at 45 DAA; Red circle—exceptional genotypes to the relationship between plant visual scale and grain moisture and yield.

**Figure 3 ijms-23-15897-f003:**
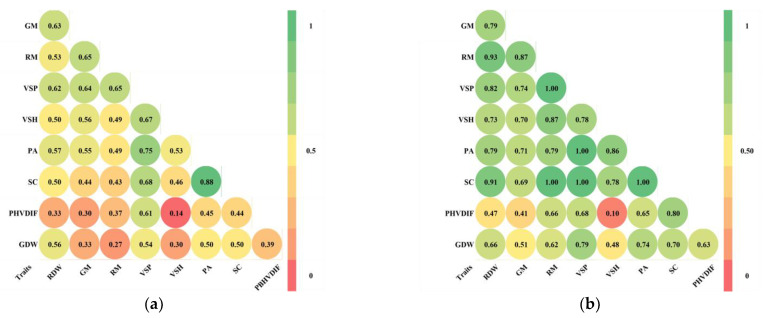
Correlation of traits with significance of *p* > 0.05. (**a**) Phenotypic correlation; (**b**) genotypic correlation. Trait: VSP—visual scale of the plant at 45 DAA; VSH—visual scale of husk leaves at 45 DAA; PHVDIF—the difference between the visual scale of plant and husk at 45 DAA; PA—photosynthesis activity at 45 DAA; SC—stomatal conductance at 45 DAA; GDW—grain dry weight; RDW—residual dry weight; GM—grain moisture; RM—residual moisture.

**Figure 4 ijms-23-15897-f004:**
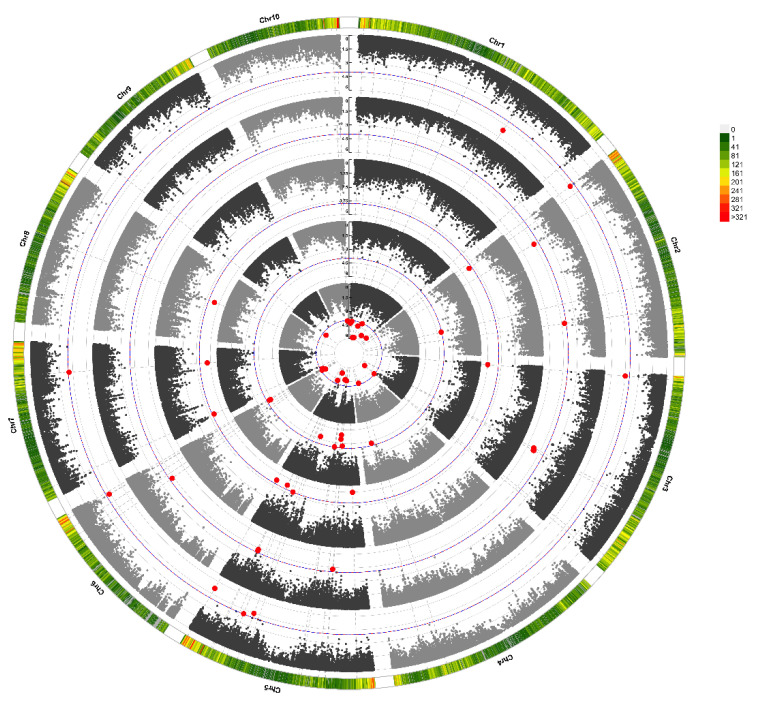
Circular Manhattan Plot. Red dots are significant SNPs for corresponding senescence traits. From inside to out: VSH—visual scale of husk leaves at 45 DAA; VSP—visual scale of plant at 45 DAA; SC—stomatal conductance; PHVDIF—the difference between the visual scale of plant and husk at 45 DAA; PA—photosynthesis activity at 45 DAA.

**Figure 5 ijms-23-15897-f005:**
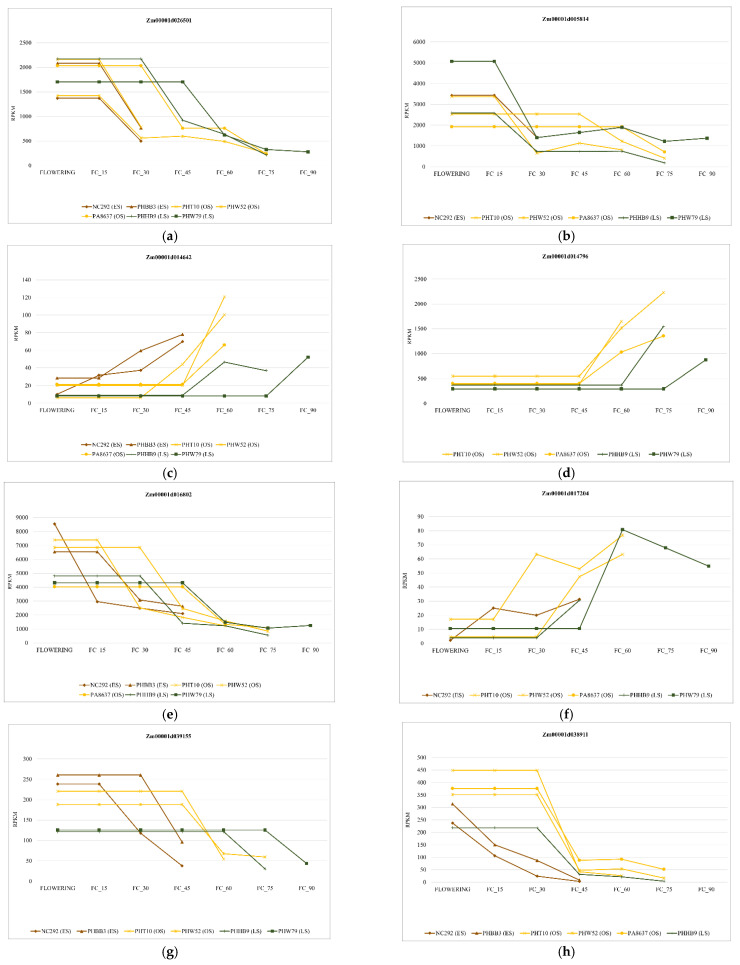
Gene expression levels during senescence, data from [20]. FC_15—RNA-seq performed 15 days after flowering (RPKM—reads per kilobase million); FC_30—RNA-seq performed 30 days after flowering (RPKM—reads per kilobase million); FC_45—RNA-seq performed 45 days after flowering (RPKM—reads per kilobase million); FC_60—RNA-seq performed 60 days after flowering (RPKM—reads per kilobase million); FC_75—RNA-seq performed 75 days after flowering (RPKM—reads per kilobase million); FC_90—RNA-seq performed 90 days after flowering (RPKM—reads per kilobase million). PHBB3, NC292—early senescence genotypes (ES). PHT10, PHW52, PA8637—optimum senescence genotypes (OS). PHBB9, PHW79—late senescence genotypes (LS). (**a**) Zm00001d026501; (**b**) Zm00001d005814; (**c**) Zm00001d014642; (**d**) Zm00001d014796; (**e**) Zm00001d016802; (**f**) Zm00001d017204; (**g**) Zm00001d039155; (**h**) Zm00001d038911.

**Table 1 ijms-23-15897-t001:** Trait performance across years.

Trait	VG ^1^	VGE ^2^	VE ^3^	Years ^4^	h^2^	Min ^5^	Max ^6^	Mean ± SE ^7^	SD ^8^	CV ^9^
VSP	1.1	0.08	0.48	3	0.85	1	5	2.52 ± 0.03	1.29	51.26
VSH	0.55	0.19	0.39	3	0.74	1	5	1.58 ± 0.02	1.04	65.82
PHVDIF	0.37	0.11	0.49	3	0.65	1	5	0.93 ± 0.02	0.99	106.19
PA	22.9	17.14	20.56	3	0.65	0	37.8	6.8 ± 0.23	8.58	126.07
SC	0.000628	0.000368	0.000945	3	0.59	0	0.32	0.03 ± 0.001	0.05	136.49
GM	1334.46	249.81	230.16	2	0.85	79.7	312	186.06 ± 1.57	49.10	26.39
GDW	260.35	117.05	226.71	2	0.6	7.72	123	53.3 ± 0.81	25.40	47.61
RM	3840.5	4671.11	6263.09	2	0.41	91.4	788	520 ± 4.26	143.98	27.68
RDW	395.33	162.58	201.62	2	0.69	9.62	161	60.14 ± 0.96	30.04	49.94

^1^ VG—variance of genotype; ^2^ VGE—the interaction of genotype and year variance; ^3^ VE—variance of residual; ^4^ Years—number of years; ^5^ Min.—minimum value; ^6^ Max—maximum value; ^7^ SE—standard error; ^8^ SD—standard deviation; ^9^ CV—coefficient of variance; h^2^—heritability of trait; Trait: VSP—visual scale of a plant (1–5) at 45 DAA; VSH—visual scale of bracts (1–5) at 45 DAA; PHVDIF—the difference between the visual scale of plant and husk at 45 DAA; PA—photosynthesis activity (mol CO_2_ m^−2^ s^−1^) at 45 DAA; SC—stomatal conductance (mol H_2_O m^−2^ s^−1^) at 45 DAA; GM—grain moisture (g/Kg of the total sample); GDW—grain dry weight (g/plant); RM—residual moisture (g/Kg total sample); RDW—residual dry weight (g/plant).

**Table 2 ijms-23-15897-t002:** List of SNPs related to senescence traits in comparison with biomass traits.

S. No	Marker ^1^	Chr ^2^	Site ^3^	Allele	Sen				BIOMASS							
									GM		RM		GDW		RDW	
					Trait ^4^	Sig ^5^	Eff ^6^	R^2^ (%) ^7^	Sig ^5^	Eff ^6^	Sig ^5^	Eff ^6^	Sig ^5^	Eff ^6^	Sig ^5^	Eff ^6^
1	S1_100824497	1	103,203,345	C/T	VSH	3.32 × 10^−5^	−0.54	5.11	1.71 × 10^−3^	−18.80	3.35 × 10^−3^	−42.18	1.16 × 10^−2^	−7.69	8.18 × 10^−3^	−9.24
2	S1_100903343	1	103,279,958	C/T	VSH	1.63 × 10^−6^	−0.67	7.50	1.89 × 10^−3^	−19.79	1.54 × 10^−3^	−48.29	2.37 × 10^−2^	−7.29	2.49 × 10^−3^	−11.21
3	S1_141282152	1	143,508,124	C/T	VSH	8.79 × 10^−5^	−0.53	4.32	7.16 × 10^−3^	−16.73	3.62 × 10^−2^	−31.01	1.17 × 10^−1^	−4.84	1.14 × 10^−2^	−8.94
4	S1_209120152	1	212,105,133	C/G	VSH	5.80 × 10^−6^	0.79	5.84	3.86 × 10^−3^	22.88	5.49 × 10^−2^	36.59	1.04 × 10^−1^	6.59	4.80 × 10^−4^	16.33
					PA	7.15 × 10^−6^	5.41	5.75								
5	S1_22259426	1	22,612,531	C/G	VSH	7.98 × 10^−5^	−0.61	4.66	1.50 × 10^−1^	−10.05	7.76 × 10^−1^	−4.81	1.27 × 10^−1^	−5.48	1.78 × 10^−1^	−5.52
6	S1_296203927	1	301,677,536	C/T	VSH	8.21 × 10^−6^	0.73	6.18	1.71 × 10^−3^	23.14	2.61 × 10^−1^	20.24	6.11 × 10^−2^	7.05	4.37 × 10^−3^	12.28
7	S1_63150814	1	63,980,587	C/G	VSH	1.45 × 10^−6^	−1.16	6.87	8.87 × 10^−4^	−36.13	3.99 × 10^−2^	−53.77	1.23 × 10^−1^	−8.75	4.71 × 10^−3^	−18.45
8	S1_6423401	1	6,420,128	A/C	VSH	5.05 × 10^−5^	0.91	5.03	9.46 × 10^−2^	16.81	7.62 × 10^−1^	−7.42	2.76 × 10^−1^	5.68	2.30 × 10^−1^	7.14
9	S2_13299973	2	13,592,073	A/C	SC	5.56 × 10^−5^	0.03	4.85	1.17 × 10^−1^	16.33	3.27 × 10^−2^	53.83	1.07 × 10^−1^	8.85	4.11 × 10^−3^	18.20
10	S2_152505815	2	157,225,727	A/G	VSP	8.92 × 10^−5^	0.61	4.34	5.51 × 10^−3^	15.09	6.05 × 10^−4^	46.11	4.30 × 10^−2^	5.71	2.75 × 10^−3^	9.69
11	S2_184012260	2	189,518,660	C/G	PHVDIF	6.45 × 10^−5^	−0.46	5.29	2.03 × 10^−1^	−7.86	2.05 × 10^−2^	−34.66	6.02 × 10^−1^	1.61	1.04 × 10^−1^	−5.74
12	S2_2088378	2	2,085,417	C/G	PA	3.43 × 10^−5^	−3.33	5.59	8.89 × 10^−2^	−9.28	2.79 × 10^−3^	−39.55	1.34 × 10^−2^	−6.94	2.19 × 10^−3^	−9.87
13	S2_42234632	2	44,077,439	A/G	PHVDIF	3.21 × 10^−5^	−0.65	5.22	1.54 × 10^−2^	−19.51	1.89 × 10^−2^	−46.97	2.25 × 10^−2^	−9.50	4.14 × 10^−4^	−16.88
14	S3_148031626	3	149,339,963	C/T	PHVDIF	5.25 × 10^−6^	0.68	6.70	2.53 × 10^−2^	17.76	2.56 × 10^−1^	21.75	5.01 × 10^−2^	7.96	1.16 × 10^−1^	7.33
15	S3_151970626	3	153,385,457	A/G	PHVDIF	6.27 × 10^−6^	0.44	6.19	1.02 × 10^−2^	13.24	1.39 × 10^−1^	18.53	8.31 × 10^−2^	4.56	3.27 × 10^−1^	2.94
16	S3_217840430	3	221,449,552	G/T	VSH	3.73 × 10^−6^	−1.04	7.38	5.42 × 10^−3^	−27.78	4.54 × 10^−4^	−87.79	5.13 × 10^−1^	−3.44	8.70 × 10^−1^	0.98
17	S3_223310403	3	227,056,710	A/C	VSH	8.54 × 10^−5^	−0.54	5.54	5.63 × 10^−2^	−11.91	4.63 × 10^−1^	−11.17	2.64 × 10^−1^	−3.58	3.66 × 10^−1^	−3.29
18	S3_5884845	3	5,057,742	C/T	PA	3.45 × 10^−5^	−3.42	5.39	1.32 × 10^−4^	−21.65	3.25 × 10^−3^	−40.11	5.01 × 10^−2^	−5.52	5.46 × 10^−3^	−8.99
					SC	1.07 × 10^−5^	−0.02	6.01								
19	S4_181727580	4	184,669,210	A/C	VSH	6.93 × 10^−5^	−0.48	4.91	2.36 × 10^−2^	−12.46	3.65 × 10^−3^	−38.16	3.38 × 10^−2^	−5.87	4.60 × 10^−2^	−6.34
20	S4_202887002	4	207,316,178	A/T	VSP	5.65 × 10^−5^	1.20	4.86	7.34 × 10^−3^	27.74	4.95 × 10^−3^	72.45	1.29 × 10^−2^	13.28	2.09 × 10^−3^	18.73
21	S5_154600488	5	158,376,423	A/G	PA	3.36 × 10^−5^	−4.15	5.45	3.84 × 10^−2^	−15.06	1.31 × 10^−2^	−42.16	3.28 × 10^−3^	−10.55	2.39 × 10^−2^	−9.32
22	S5_147166110	5	150,672,499	C/T	VSP	1.69 × 10^−5^	−0.85	5.97	1.12 × 10^−3^	−22.69	3.55 × 10^−3^	−47.37	5.24 × 10^−2^	−6.61	2.48 × 10^−2^	−8.81
					VSH	4.83 × 10^−6^	−0.69	6.76								
23	S5_167009258	5	170,916,787	C/T	PA	9.04 × 10^−5^	−6.48	4.78	3.87 × 10^−2^	−22.47	4.76 × 10^−2^	−53.38	9.19 × 10^−2^	−9.48	3.86 × 10^−2^	−13.27
					SC	9.83 × 10^−5^	−0.04	4.67								
24	S5_172396840	5	176,501,182	C/G	VSH	4.28 × 10^−5^	0.61	4.86	2.19 × 10^−1^	8.42	3.28 × 10^−2^	35.03	8.99 × 10^−1^	0.43	1.22 × 10^−1^	6.04
25	S5_184169070	5	188,797,787	A/C	PHVDIF	6.45 × 10^−5^	−0.43	5.03	1.31 × 10^−1^	−8.44	8.09 × 10^−3^	−36.42	3.63 × 10^−1^	−2.61	1.19 × 10^−1^	−5.10
26	S5_184171751	5	188,800,468	A/T	PHVDIF	4.64 × 10^−5^	0.47	5.20	2.64 × 10^−1^	6.79	1.35 × 10^−1^	21.94	1.91 × 10^−1^	4.03	1.26 × 10^−1^	5.40
27	S5_185866560	5	190,535,974	A/G	SC	4.12 × 10^−5^	0.03	4.94	1.51 × 10^−2^	20.31	5.01 × 10^−2^	40.18	1.71 × 10^−1^	5.96	2.50 × 10^−1^	5.73
28	S5_19709941	5	20,195,993	C/T	SC	1.06 × 10^−5^	−0.02	5.81	1.40 × 10^−1^	−7.65	1.47 × 10^−1^	−18.29	7.24 × 10^−2^	−4.79	3.67 × 10^−2^	−6.38
29	S5_214537055	5	220,409,338	A/G	PA	6.97 × 10^−6^	−3.80	6.49	1.74 × 10^−2^	−13.74	1.40 × 10^−1^	−20.39	5.35 × 10^−3^	−8.06	1.31 × 10^−3^	−10.66
					SC	4.56 × 10^−5^	−0.02	5.22								
30	S5_54415632	5	56,109,705	G/T	VSP	6.22 × 10^−5^	0.89	4.74	7.52 × 10^−4^	26.26	1.64 × 10^−4^	71.22	7.50 × 10^−2^	7.09	1.02 × 10^−3^	15.10
31	S5_55394662	5	57,153,911	A/C	PHVDIF	5.08 × 10^−5^	0.60	4.84	4.90 × 10^−2^	15.55	7.30 × 10^−3^	51.32	5.80 × 10^−1^	2.21	2.18 × 10^−1^	5.63
32	S5_61802239	5	63,587,555	C/T	VSH	2.68 × 10^−5^	0.74	5.37	3.81 × 10^−3^	23.31	1.15 × 10^−2^	48.79	1.15 × 10^−2^	10.03	4.89 × 10^−2^	8.96
					VSP	3.78 × 10^−6^	1.06	6.66								
33	S5_80969487	5	83,171,395	C/G	VSH	2.27 × 10^−5^	0.53	5.83	1.64 × 10^−2^	13.55	1.77 × 10^−2^	32.46	9.38 × 10^−3^	7.48	2.05 × 10^−3^	10.17
34	S5_84063981	5	86,287,916	C/T	VSP	9.09 × 10^−5^	0.70	4.38	4.35 × 10^−3^	18.00	1.86 × 10^−3^	46.90	4.09 × 10^−3^	9.05	4.77 × 10^−4^	12.69
35	S6_132457507	6	136,597,839	G/T	PHVDIF	5.17 × 10^−5^	−0.47	5.24	2.47 × 10^−3^	−18.90	2.91 × 10^−3^	−44.78	3.32 × 10^−2^	−6.73	2.10 × 10^−3^	−11.14
36	S6_141797374	6	145,913,312	C/T	VSH	3.08 × 10^−5^	0.98	5.35	7.53 × 10^−3^	28.34	2.91 × 10^−3^	77.85	2.77 × 10^−1^	5.93	8.92 × 10^−3^	16.30
37	S6_152309276	6	156,446,826	A/G	VSH	7.63 × 10^−5^	0.57	5.25	1.24 × 10^−1^	10.01	2.55 × 10^−1^	18.05	2.21 × 10^−2^	7.74	1.18 × 10^−2^	9.68
38	S6_162646831	6	166,851,405	C/G	VSP	3.76 × 10^−5^	0.67	5.37	4.32 × 10^−4^	20.14	1.29 × 10^−3^	43.93	1.31 × 10^−2^	7.15	3.40 × 10^−5^	13.83
39	S6_163349566	6	167,553,244	C/G	PA	4.42 × 10^−5^	−3.11	4.77	9.63 × 10^−4^	−17.46	5.71 × 10^−2^	−23.98	4.00 × 10^−4^	−9.33	3.19 × 10^−3^	−8.86
					VSP	5.26 × 10^−5^	−0.61	4.78								
					VSH	4.88 × 10^−5^	−0.47	4.73								
40	S6_167031439	6	171,178,929	C/T	VSH	6.13 × 10^−5^	0.87	5.41	2.51 × 10^−1^	11.19	8.05 × 10^−1^	5.90	3.30 × 10^−1^	5.00	2.69 × 10^−2^	12.95
41	S7_12615372	7	13,041,805	A/G	SC	6.97 × 10^−5^	0.03	4.38	3.54 × 10^−1^	9.72	3.68 × 10^−1^	22.66	7.07 × 10^−2^	8.67	4.21 × 10^−1^	4.38
42	S7_139201593	7	143,942,346	C/T	SC	1.99 × 10^−5^	0.04	5.04	9.69 × 10^−2^	18.18	4.29 × 10^−3^	77.67	1.10 × 10^−1^	9.23	7.35 × 10^−3^	17.76
					PA	8.37 × 10^−5^	6.44	4.26								
43	S8_96765835	8	98,565,103	A/G	SC	2.89 × 10^−5^	0.04	6.11	4.84 × 10^−1^	8.28	2.21 × 10^−2^	66.36	1.85 × 10^−2^	13.24	2.43 × 10^−2^	14.42
44	S9_8660567	9	8,358,103	A/G	VSH	4.06 × 10^−5^	0.59	5.13	9.42 × 10^−2^	10.87	4.73 × 10^−2^	31.13	4.30 × 10^−1^	2.59	4.54 × 10^−1^	2.81
45	S10_141078749	10	141,806,270	G/T	VSH	8.94 × 10^−5^	0.47	4.60	7.28 × 10^−2^	9.70	5.21 × 10^−2^	25.80	3.89 × 10^−1^	2.37	2.31 × 10^−1^	3.77
46	S10_146472807	10	147,246,775	G/T	VSH	8.56 × 10^−5^	−0.53	4.63	1.68 × 10^−4^	−23.51	5.97 × 10^−3^	−40.65	8.11 × 10^−1^	−0.70	1.66 × 10^−3^	−10.67

^1^ Marker—significant SNPs for the desired trait; ^2^ Chr—physical chromosomal position of the marker; ^3^ Site—physical position of the SNP in bp; ^4^ Trait-GM—grain moisture; RM—residual moisture; GDW—grain dry weight; residual dry weight; VSP—visual scale of the plant at 45 DAA; VSH—visual scale of husk at 45 DAA; PHVDIF—the difference between the visual scale of plant and husk at 45 DAA; PA—photosynthesis activity at 45 DAA; SC—stomatal conductance at 45 DAA; ^5^ SIG—significance; ^6^ EFF—effect; ^7^ R^2^ proportion of the phenotypic variance explained by the QTL.

**Table 3 ijms-23-15897-t003:** List of candidate genes with significant markers within the coding region of the gene and in the promoter region.

S. No	Marker ^1^	Trait ^2^	Position ^3^	Gene ID ^4^	Start ^5^	Stop ^6^	Gene location ^7^	Description
1	S2_184012260	PHVDIF	189,518,660	Zm00001d005814 **	189,518,235	189,520,622	CR	Lhca6
2	S5_55394662	PHVDIF	57,153,911	Zm00001d014642 *	57,152,592	57,154,619	CR	EXO70
3	S5_61802239	VSP	63,587,555	Zm00001d014796 *	63,586,418	63,588,955	CR	Uncharacterized protein
		VSH						
4	S5_172396840	VSH	176,501,182	Zm00001d016802 **	176,496,325	176,501,360	CR	Ascorbate peroxidase (APX)
5	S5_184169070	PHVDIF	188,797,787	Zm00001d017204 *	188,797,631	188,801,896	CR	Adenine phosphoribosyltransferase 2
6	S6_162646831	VSP	166,851,405	Zm00001d038911 **	166,851,524	166,851,829	119 bp PR	Nonspecific lipid-transfer protein
7	S6_167031439	VSH	171,178,929	Zm00001d039155 **	171,177,893	171,180,993	CR	Tudor/PWWP/MBT superfamily protein
8	S10_146472807	VSH	147,246,775	Zm00001d026501 **	147,240,629	147,249,381	CR	Glutamine synthetase%2C chloroplastic

^1^ Marker—significant SNP; ^2^ Trait—PHVDIF, VSP, VSH. ^3^ Position—physical position of SNP in bp; ^4^ Gene ID—gene id asper B73 refgen_v4; * gene upregulation during senescence; ** gene downregulation during senescence; ^5^ Start—gene starting position in bp; ^6^ Stop—gene stop position in bp; ^7^ Description—general description of the gene (CR—coding region; PR—promoter region).

## Data Availability

Not applicable.

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
