# Peer review of "Genome-Wide Association Analysis of Senescence-Related Traits in Maize"

_ijms, 2022, doi:10.3390/ijms232415897_

Round 1
Reviewer 1 Report
Conclusions should not include references or citations, it should be the original work conclusion only.
A clear-cut result and analysis of work are lacking.
The outcome of work should be justified strongly which is lacking.
The way forward of the research is lacking/missing.
Overall, the work bears merit.
Author Response
We would like to thank you for giving us a chance for revision of the manuscript according to the reviewer’s comments. We would like to thank all the reviewers, for their thoughtful comments and efforts toward improving the quality of our manuscript.
We have acquired the formatting and English service recommended by the editor of MDPI. The quality of the figures and the serial number of the references are updated according to their recommendation.
We follow the recommendations of the reviewers.
Answer to reviewer1
We modified the conclusions following the indications of reviewer 1. We deleted the citations for the conclusions. We reduced the conclusions to highlight the main aspects of the design and the key results obtained. The two main results that we remarked on were: first the high correlations and the coincidence of the allelic effects among senescence and agronomic traits which imply that senescence is an important trait for adaptation, and second, the identification of eight genes by two independent studies (association and RNAseq) that make them strong candidates for subsequent functional studies and breeding. With those modifications in the conclusions, we expect that now that “clear-cut results and analysis”, a “strong justification of the outcome” and a “way forward of the research” will be clear for the readers of the manuscript as the reviewer requested. According to your suggestion for Moderate English changes required, with the help of an English editing service and formatting service from MDPI the manuscript was undergone by the specialist/native English speaker. In the same way, the figures quality and the serial number of the references are updated according to the recommendation of the layout service.
Once again we would like to thank you for giving us another chance. I hope we addressed all the remarks of the reviewers.
Thanks and Regards

Reviewer 2 Report
Excellent research. But it would be better if in its introduction it integrates the hypothesis or research questions.
The introduction does not present the research question. It is the recommendation to improve; the authors must integrate the research question or research hypothesis addressed.I consider the topic original or relevant in the field
Identification of candidate genes for senescence and the evaluation of senescence in peels independently of the remaining leaves.
The conclusions are consistent according to the objectives set. But it is not possible to identify whether the consistency of the research regarding the research question, because it is not presented.
References are appropriate.
The research is very relevant and the results are a good contribution to knowledge on the subject.
Author Response
We would like to thank edit for giving us a chance for revision the manuscript according to the reviewer’s comments. We would like to thank all the reviewers, for their thoughtful comments and efforts toward improving the quality of our manuscript.
We have acquired the formatting and English service recommended by the editor of MDPI. The quality of the figures and the serial number of the references are updated according to their recommendation.
We follow your recommendations
Answer to reviewer2
We rewrote the objectives of the research, and we think that they are now more clear for the reader. According to your suggestion, minor spell checks are required, with the help of an English editing service and formatting service from MDPI the manuscript was undergone by a specialist/native English speaker. In the same way, the figure's quality and the serial number of the references are updated according to the recommendation of the layout service.
Once again we would like to thank you for giving us another chance. I hope we addressed all the remarks of the reviewers.
Thanks and Regards